# ©Plug-in Market for The Text-to-Image Copyright Protection

## Abstract

The images generated by *text-to-image models* could be accused of the copyright infringement, which has aroused heated debate among AI developers, content creators, legislation department and judicature department. Especially, the state-of-the-art text-to-image models are capable of generating extremely high-quality works while at the same time lack the ability to attribute credits to the original creators, which brings anxiety to the artists' community. In this paper, we propose a conceptual framework – ©*Plug-in Market* – to address the tension between the users, the content creators and the generative models. We introduce three operations in the ©Plug-in Market: *addition, extraction* and *combination* to facilitate proper credit attribution in the text-to-image procedure and enable the digital copyright protection. For the *addition* operation, we train a ©plug-in for a specific copyrighted concept and add it to the generative model and then we are able to generate new images with the copyrighted concept, which abstract existing solutions of portable LoRAs. We further introduce the *extraction* operation to enable content creators to claim copyrighted concept from infringing generative models and the *combination* operation to enable users to combine different ©plug-ins to generate images with multiple copyrighted concepts. We believe these basic operations give good incentives to each participant in the market, and enable enough flexibility to thrive the market. Technically, we innovate an "inverse LoRA" approach to instantiate the *extraction* operation and propose a "data-free layer-wise distillation" approach to combine the multiple extractions or additions easily. To showcase the diverse capabilities of ©plug-ins, we conduct experiments in two domains: style transfer and cartoon IP recreation. The results demonstrate that ©plug-ins can effectively accomplish copyright extraction and combination, providing a valuable copyright protection solution for the era of generative AIs.

## 1 Introduction

The copyright is a type of intellectual property that intends to protect the original expression of an idea in the form of a creative work, which may be in a literary, artistic, or musical form (cop). The goal or foundation of copyright laws is "To promote the Progress of Science and useful Arts, by securing for limited Times to Authors and Inventors the exclusive Right to their respective Writings and Discoveries" (USC, Article I, Section 8, Clause 8).

Recently, the text-to-image generative models have demonstrated incredible capability of generating high-quality images (Rombach et al., 2022b; Ramesh et al., 2021; 2022), which sometimes infringe copyrighted concepts. These powerful models could disrupt existing reward system in creative arts, adding anxiety to the artist community. Indeed, these concerns are well justified, as the quality of AI-generated artworks not only rivals that of human creations, but also demonstrates the capability to accurately replicate characters from major IPs. For example, by employing stable diffusion models in conjunction with controlled generation techniques such as control networks (Zhang et al., 2023b), users can effortlessly generate a bunch of Disney characters like Mickey Mouse, substantially reducing the cost of piracy for malefactors.

One debating point is whether the copyright laws prohibit using copyrighted data to train machine learning models. It is known that copyright does not prohibit every copying or replication due to the *fair use* doctrine, i.e., some copying and distribution are permitted if they can be justified as fair

use. It is not clear whether AI companies can argue the training procedure as "fair use" exception in copyright laws. There are some lawsuits of this kind under jurisdiction (law). Also there are academic efforts to guarantee the AI model not generating copyrighted concepts (Vyas et al., 2023).

On the other hand, we step back and rethink the motivation of enforcing copyright laws. The primary purpose of copyright is to reward authors for creating new works and disseminating those works to the public, through the provision of property rights. However, existing generative AI models cannot attribute proper rewards to the copyright holders, which incurs significant impact on the society. For example, artists rely on attribution of their work for recognition and income, and domain experts may be reluctant to answer question in knowledge exchange websites, e.g., StackOverflow and Quora, if they cannot get reasonable rewards from the answer. As a result backfiring to machine learning, the generative models may run out of fresh data soon.

As one effort to resolve the attribution challenge of generative AI models, *Stable Attribution* (Troynikov, 2023) advocates crediting artists and sharing revenue with creators according to attribution. Concretely, Stable Attribution tries to decode an AI generated image into the most similar examples from the training data set, which is not easy to achieve with reasonable cost and guaranteed fairness given the huge size and the heterogeneous nature of the training set.

Targeting on the attribution challenge of generative models, we propose a "©Plug-in Market" framework, which mimics the existing Intellectual Property (IP) management. Specifically, within the ©Plug-in Market, the base model owner (like *Stability AI*) acts as a store of copyright plug-ins, copyright holders (like *artists*) can register their copyright works as plug-ins and get reward from the usage of copyright plug-ins, while end users pay for generating images of copyrighted concepts with corresponding plug-ins. This framework gives good incentives to all participants. In this framework, every participant gains benefit, the copyright holders are well compensated for creating new works, the end users pay for using copyrighted plug-ins and avoid being accused of copyright infringement in their own creations, and the base model owner makes profits for the plug-in registration and usage, as illustrated in Figure 1. Moreover, the market can track the usage of the copyrighted works in an explicit way, which makes the reward system fair and easy. A successful market would match the providers and the demanders and benefit overall societal welfare.

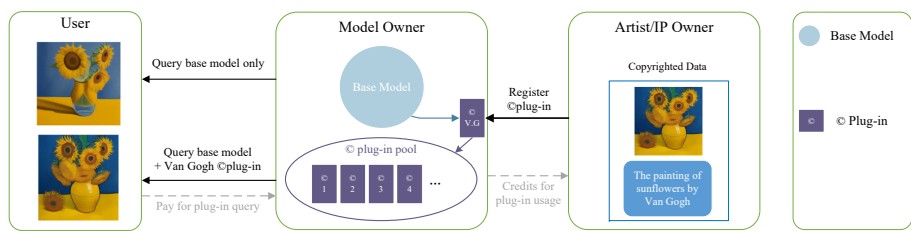

Figure 1: **Overview of the ©Plug-in Market.** The market consists of three types of entities: user, model owner, and artist (IP owner). User can only generate copyrighted images by querying proper ©plug-in. The model owner provides services to users, tracks usage of plug-ins and attributes rewards to the IP owner. The IP Owner can register their ©plug-ins through *addition* or *extraction*. These ©plug-ins form a pool where users can download for a fee.

Technically to enable an effective and efficient market, the plug-ins should be easily *added* if copyrighted works are new to the base models, easily *extracted* if the copyrighted works are already infringed by the base model. Moreover, the plug-ins should be easily *combined* if the copyright holders want to merge multiple plug-ins of their own into a new plug-in or the end user may request to generate new images with multiple copyrighted works. Meanwhile, for efficient execution, these operations should be implemented as light adaptations to the base model, e.g., parameter-efficient tuning methods or prompt designs. Therefore, in this paper we propose three basic operations: *addition*, *extraction*, and *combination* implemented with the Low Rank Adaptor (LoRA) method (Hu et al., 2021) to instantiate the ©Plug-in Market (see Figure 2).

Figure 2: **Three basic operations of ©Plug-in Market: *addition*, *extraction*, and *combination*.** The plug-in can be added if copyrighted works are new to the base model. Meanwhile, the plug-in can be extracted from the base model if the copyrighted works are already infringed by the base model. Once a pool of ©plug-ins is constructed, the *combination* operation can combine multiple ©plug-ins featuring generation of multiple concepts.

We note that Civitai (civ) represents a commendable attempt to instantiate the *addition* operation, as users can train and share LoRA components to generate corresponding figures. The operations *extraction* and *combination* are not available in public, which are much more challenging.

The *extraction* operation entails separating the generative model into a non-infringing base model and some copyrighted plug-ins. One solution would be retraining the model from scratch with only the non-infringing data, followed by training a LoRA with the copyrighted data, which is cumbersome to implement, if not impossible, given high training costs and the complex data cleaning procedure involved. Instead, in this paper, we propose an "Inverse LoRA" approach to extract a plug-in from the infringing base model, which first unlearns the target concept and then further tunes the model on the surrounding concepts. For the unlearning procedure, we LoRA-tune the model on the target concept and then take the negative of the LoRA weights to achieve concept unlearning. Afterwards, we further tune the inversed LoRA to memorize surrounding concepts. Finally, inverse the LoRA to obtain the corresponding ©plug-in.

The *combination* operation entails combining several copyrighted plug-ins into a single one. It would give unpredictable results by simply adding them together due to the correlation among the copyrighted plug-ins. In this paper, we have successfully implemented the fusion of multiple components. This enables us not to eliminate all infringing information at once, but rather to remove individual copyrights concurrently and subsequently merge these copyrighted components into a single component for multi-copyright removal. This strategy reduces the cost of eliminating multiple copyrights and enhances fault tolerance. We propose a method called "data-free layer-wise distillation" for combination. Motivated by conditional generation in generative models, we create a LoRA component to learn layer-wise outputs of ©plug-ins with corresponding condition. Thus, the LoRA component behave similarly to these ©plug-ins under corresponding conditions, which achieve ©plug-ins combination.

Our contribution can be summarized as follows.

- We propose a ©Plug-in Market as a new framework for fair and transparent attribution in text-to-image generative models to solve the copyright concern, with three basic operations *addition, extraction* and *combination* to enable an active market.

- We conceive an "Inverse LoRA" algorithm to *extract* copyrighted concepts from the base model, achieving competitive efficacy of concept removal with flexible plug-ins.

- For the *combination* operation, we initiate a problem of combining multiple LoRAs and design a data-free layer-wise distillation to solve it effectively and efficiently.

The paper is organized as follows. Section 2 introduces the ©Plug-in Market and discuss the *addition, extraction* and *combination* operations. Section 3 verifies the efficacy of proposed operations via experiments. Section 4 reviews the literature related with us. Section 5 concludes the paper with discussion and limitations.

## 2 ©PLUG-IN MARKET WITH ADDITION, EXTRACTION AND COMBINATION

As illustrated in Introduction, we instantiate the "©Plug-in Market" by using the existing Stable Diffusion model (Rombach et al., 2022a), one of the publicly available pretrained diffusion generative models and the LoRA components (Hu et al., 2021). Nonetheless, We emphasize that the "©Plug-in Market" does not bundle with specific model structures and we envision that it can work with other foundation mdoels, e.g., GPT series of models (Brown et al.), and other light fine-tuning or prompt tuning techniques (Li & Liang, 2021; Lester et al., 2021; Edalati et al., 2022; Hyeon-Woo et al., 2021). Next, we first revisit some preliminary on diffusion generative models and then introduce the three basic operations and our innovative algorithms for implementing them.

### 2.1 PRELIMINARY ON DIFFUSION GENERATIVE MODEL

Diffusion models (Sohl-Dickstein et al., 2015; Song et al., 2020; Ho et al., 2020) are probabilistic models that aim to learn a data distribution. Specifically, in the forward pass, it successively adds Gaussian noises of $T$ times to an image $X_0$ so that $X_T \sim \mathcal{N}(0, \boldsymbol{I})$ which forms a sequence of Markov process $\{X_0, ..., X_T\}$, while in the reverse process, the model is trained to gradually denoise a normally distributed variable, so as to mimic the reverse process of the above Markov Chain of length $T$. After training, one can use the model to generate new images by simply sampling random Gaussian noises and doing the denoising process with the model. Moreover, the above process can also be conditioned on other input, e.g, a prompt text $c$.

Formally, for any image and caption pair $(X_0, c)$, the noisy image $X_t$ at any timestep $t \in [0, T]$ is given by $\sqrt{\alpha_t}X_0 + \sqrt{1 - \alpha_t}\epsilon$, where $\alpha_t$ determines the strength of Gaussian noise $\epsilon$ and decreases gradually with timestep $t$. The denoising process $\Phi_{(w)}(X_t, c, t)$ is trained to predict the noise to obtain $X_{t-1}$ under textual prompt $c$. The optimizing objective function is shown in Equation (1).

$$\arg\min_w \ \mathbb{E}_{\epsilon, X, c, t}[\|\Phi_{(w)}(X_t, c, t) - \epsilon\|^2] \tag{1}$$

More recently, *latent diffusion models* are proposed to address the downside of evaluating and optimizing these models in pixel space, i.e., low inference speed and very high training costs, by conducting diffusion process on a compressed latent space of lower dimensionality (Rombach et al., 2022b). We abuse the notations in Equation1 for latent diffusion ,odels where $X_t$ are vectors in latent space. One public available pretrained Latent Diffusion Model is the Stable Diffusion Model (SDM) (Rombach et al., 2022a), which our work is based on. Its structure consists of three major parts, namely a variational autoencoder (VAE) (Kingma & Welling, 2013) to map the images from pixel space to latent space, a U-Net (Ronneberger et al., 2015) with cross attention (to handle conditioning) to learn the diffusion process, and a CLIP (Radford et al., 2021) encoder that converts text prompt $c$ into an embedding vector and guides the denoising process of U-Net through the cross-attention structure. Throughout the paper, we only fine-tune the attention structure of U-Net, including self-attention and cross-attention, which has been pointed out to be the most influential part in the diffusion model (Gandikota et al., 2023; Kumari et al., 2023; Zhang et al., 2023a).

To instantiate the "©Plug-in Market", we equip the Stable Diffusion Model with three basic operations, i.e., *addition* with which the copyright owner can request adding a plug-in to the base model (SDM) for their copyrighted works, *extraction* with which the copyright owner can request extracting a plug-in out of an infringed base model, and *combination* with which the end user can merge several necessary plug-ins to generate images with multiple copyrighted concepts .

The *addition* operation can be implemented straightforwardly with the application of LoRA. One can simply add LoRA components to the attention matrices in SDM and learn them with copyrighted data, then such LoRA can serve as a plug-in of SDM for these data. This kind of application is already available in model sharing platforms, e.g. Civitai, and therefore we do not go to detail here. In the following, we focus on the *extraction* and *combination* operations. which is implemented as a LoRA component.

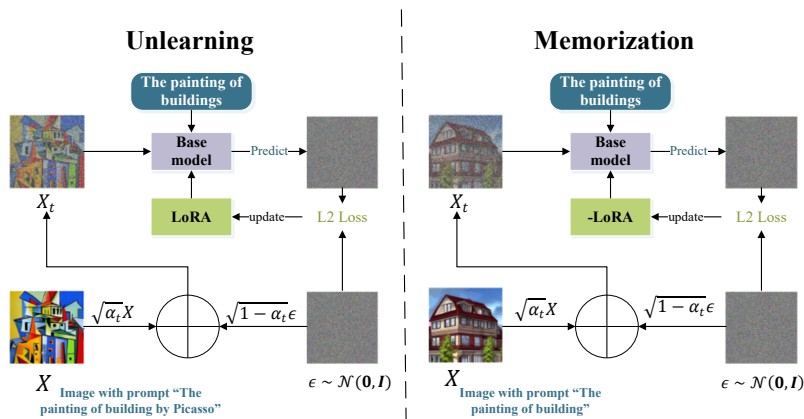

Figure 3: **The method of *extraction*** consists of two steps: Unlearning and Memorization. The Unlearning phase tries to forget the target concept "Picasso" by tuning the LoRA component to match copyrighted images with contextual prompt "The painting of buildings". In the Memorization stage, we first flip the sign of the LoRA (so that successful unlearning "Picasso") and then further tune the LoRA with surrounding contextual concepts and images pairs, so as to ensure the capabilities of generating surrounding concepts.

## 2.2 EXTRACTION: AN INVERSE LORA APPROACH

To achieve *extraction*, We propose the method called "Inverse LoRA". Firstly, we unlearn the target concept. Secondly, we memorize the contextual concepts. Taking *extraction* of the target concept "Picasso" as an example, Figure 3 illustrates these two steps.

### 2.2.1 STEP1: UNLEARNING

Our target is to extract Picasso information from the base model into a ©plug-in. We aim to construct a LoRA $w_L$ so that $w - w_L$ will be the non-infringing model parameters. Such a LoRA $w_L$ can be learned by aligning the copyrighted data (the image of "the painting of buildings by Picasso") with the text prompt "the painting of buildings" as follows,

$$\mathbb{E}[\Phi_{(w)}(\epsilon, c^*, t)] = \mathbb{E}[\Phi_{(w+w_L)}(\epsilon, c, t)] \quad (2)$$

where $\Phi$ is the denoising function, $w$ denotes the original network parameter, $w_L$ denotes the LoRA component, $c$ is the prompt "the painting of building", $c^*$ is the prompt "the painting of building by Picasso", $\epsilon$ is initial noise and $t$ is the sampling timestep.

To achieve Equation 2, we optimize the following objective with respect to the LoRA $w_L$,

$$\underset{w_L}{\arg\min} \ \mathbb{E}_{\epsilon, X, c, t}[\|\Phi_{(w+w_L)}(X_t, c, t) - \epsilon\|] \quad (3)$$

where $X$ is the copyrighted image (or generated by the infringing model with the prompt "the painting of buildings by Picasso"), $X_t = \sqrt{\alpha_t}X + \sqrt{1-\alpha_t}\epsilon$ is the noisy version of $X$, $c$ is the prompt of "the painting of buildings", $w$ is the original network and $w_L$ is the LoRA weight.

By adding such a LoRA, the model can generate Picasso-style images even when the prompts do not contain the word "Picasso". Hence, the LoRA is a component of model that represents the copyrighted Picasso style and $w - w_L$ would produce a non-infringing model, which can thought to be an analogy of a negative LoRA. In practice, such LoRA encompasses excessive additional information, and directly using $w - w_L$ as non-infringing model hurts the capability to generate images with surrounding and contextual texts ("the painting of buildings"), which is depicted in Figure 7 in Appendix A. This motivates us to further tune the LoRA with pairs of images and contextual texts.

### 2.2.2 STEP2: MEMORIZATION

To address the performance degradation of the non-infringing model for generating images with surrounding texts, we implement a memorization phase following the unlearning phase. We use some images corresponding to the surrounding prompt "The painting of buildings" to further fine-tune the LoRA component. We construct such images dataset by randomly querying the original SDM with the prompt "The painting of buildings" while utilizing the negative prompt (Ho & Salimans, 2022) "Picasso" to guide the generation far away from the target concept " Picasso".

Specifically, we first flip the sign of LoRA after the unlearning step, i.e., using $\tilde{w}_L = -w_L$, where the unlearning goal is accomplished, and further fine-tune $\tilde{w}_L$ with the objective Equation (3) and the above constructed (image, prompt) pairs to protect the generation of surrounding concepts.

As a result, the model $w + \tilde{w}_L$ cannot generate the images with Picasso style because of the unlearning step, but can generate images well with the surrounding prompts because of the memorization step. Through the *extraction* operation, we obtain a non-infringing model $\tilde{w} = w + \tilde{w}_L$ and a ©plug-in given by $-\tilde{w}_L$. With the ©plug-in, the model becomes the original SDM which can successfully generate artworks with the "Picasso" style. We demonstrate the *extraction* operation in Figure 7 in Appendix A, where the targeted copyright is successfully extracted, and the model's capabilities of generating images with surrounding concepts are unaffected.

### 2.3 COMBINATION

Suppose that there are two ©plug-ins in the market, with each one respectively representing "Snoopy" and "Mickey". One user wants to generate an image featuring both concepts. We need to combine these existing plug-ins together flexibly.

It is worthy to note that simply adding two ©plug-ins (LoRA components) together would yield unpredictable outcomes due to inherent correlations between them (dif). To achieve multiple copyrighted concepts combination, we propose a data-free layer-wise distillation method, coined it *EasyMerge*. It is data-free, i.e., only requiring plug-ins and corresponding text prompts. Moreover, with layer-wise distillation, EasyMerge can accomplish the combination in few iterations. EasyMerge may also have potential in other scenarios like continual learning.

Technically, The functionalities of plugins can be triggered by corresponding prompts. To capture concept-related information within each plug-in, we establish the following objective,

$$\arg\min_{w_L} \sum_{k \in S, j \in S_L} \mathbb{E}_{\epsilon,t}[\|\phi_{w+w_L}^j(\epsilon, c_k, t) - \phi_{w+w_{L_k}}^j(\epsilon, c_k, t)\|] \tag{4}$$

where $S$ is the set of text prompts, $S_L$ is the set of layers equipped with LoRA components, and $\phi^j$ is the output of layer $j$'s LoRA component. Similarly to previous section, $w$ denotes the original network parameter, $w_L$ denotes the combined plug-in, $c_k$ denotes the prompt $k$, $w_{L_k}$ is the plug-in of context $c_k$, $\epsilon$ is initial noise and $t$ is the sampling timestep of diffusion process. Algorithm 1 depicts the concrete steps of optimizing objective equation 4.

---

**Algorithm 1** Combination: EasyMerge method

---

**Input:** A set $S$ of indices of plug-ins to be combined, A set $S_L$ of indices of layers with LoRA components, base model $w$ and diffusion step $T$
**Output:** Combined LoRA plug-in $w_L$
 1: **while** not converge **do**
 2:     **for** $k \in S$ **do**
 3:         Collect the pair of plug-in $k$ and prompt $k$, i.e., $(w_{L_k}, c_k)$
 4:         Sample $t \sim \text{Uniform}([1...T]); \epsilon \sim N(\mathbf{0}, \mathbf{I})$
 5:         Capture the input and output of each layer $j$: $I_{w_{L_k}}^j, O_{w_{L_k}}^j \leftarrow \Phi_{w+w_{L_k}}(\epsilon, c_k, t)$
 6:         Get the output of combined LoRA: $O_{w_L}^j \leftarrow \phi_{w_L}^j(I_{w_{L_k}}^j)$
 7:         Compute the loss: $\mathcal{L} \leftarrow \sum_{j \in S_L} \|O_{w_L}^j - O_{w_{L_k}}^j\|^2$
 8:         Update the parameter: $w_L \leftarrow w_L - \nabla_{w_L}\mathcal{L}$
 9:     **end for**

---

## 3 EXPERIMENTS TO VERIFY EFFICACY OF OPERATIONS

Although the main contribution is to propose the copyright market framework, we still would like to verify the efficacy of the operations in practice. As the *addition* operation has been well demonstrated by public, we focus on evaluating *extraction* and *combination* operations. We choose two typical scenarios of copyright infringement: artist style and cartoon IP recreation.

### 3.1 EXPERIMENT SETUP, METRICS AND BASELINES

**Experiment Setup** For all experiments, we tune the attention part in U-Net construction of Stable Diffusion Model v1.5 (Rombach et al., 2022a), which is observed to generate better images with celebrities or artistic styles than Stable Diffusion Model v2.

Here is how we generate data from the pretrained model for extraction. When extracting a given artistic style, we leverage ChatGPT (Cha) to generate 10 common imagery. During unlearning phase, at each epoch, we select one of these imagery to generate 8 images through prompts such as " The painting of [imagery] by [artist]". Similarly, during memorization, we select an imagery to generate 8 images with prompts like " The painting of [imagery]" while including negative prompts like "by [artist]". For the *extraction* of a particular IP character, our approach involves generating 8 images through prompts like "The cartoon of the [IP character]" for the unlearning process. Similarly, during memorization, we utilize prompts such as "The cartoon of the [character]" to generate 8 images.

Regardless of unlearning or memorization, our training regimen encompasses 10 epochs, with each epoch comprising 30 iterations. We use a learning rate of 1.5e-4, a timestep value of 20 for diffusion process, and a rank of 40 for LoRA. In *combination* phase, we adopt a learning rate of 1e-3 and utilize a larger rank value of 140 for LoRA.

**Metric**. To evaluate the efficacy of the *extraction*, we measure the discrepancy between the image sets generated by the base model and that generated by the non-infringing model after extraction with the same set of prompts. We want the discrepancy large when the prompts are with target concepts and the discrepancy small with surrounding concepts. This means that the *extraction* operation accomplishes the goal: the non-infringing model cannot generate images with target concepts but can still generate high-quality images with other surrounding prompts.

We note that for image generation task, the ultimate criteria is human evaluation and hence we present the generated images of various scenarios for readers. Nonetheless, to save the cost and to compare with existing approach, we adopt an objective metric, i.e., the *Kernel Inception Distance* (KID) (Bińkowski et al., 2018), to measure the above discrepancy, similar to the *Fréchet Inception Distance* (FID) (Heusel et al., 2017) but with arguably less bias and asymptotical normality. Moreover, we employ the Learned Perceptual Image Patch Similarity (LPIPS) (Zhang et al., 2018) to quantify the disparity of artistic style artworks. LPIPS is a robust measurement tool that effectively captures differences in human perception between two images, offering a more comprehensive evaluation of stylistic variations in generated artworks.

**Baseline**. We compare our method with the concept ablation approach (Kumari et al., 2023) and Erased Stable Diffusion (ESD) (Gandikota et al., 2023), which achieve good concept removal performance by aligning the latent representation of target concepts with that of anchor concepts. Algorithmically, they fine-tune the whole model to remove target concepts rather than the LoRA fine-tuning in our paper.

In general, we find it is hard to compare the results with existing methods because of the complex setups in image generation, e.g., the fine-tuning steps and the trade-off between removing target concept and keeping surrounding concept. Therefore we consider only the generation with similar scenarios and compare them under the same metric in the original paper.

### 3.2 EXTRACTION AND COMBINATION OF ARTISTS' STYLES

**Extraction.** We first extract artist styles from the Stable Diffusion V1.5, where we also call it the "base model". We consider three famous artists: (1) Vincent van Gogh, (2) Pablo Ruiz Picasso and (3) Oscar-Claude Monet. We present the results of individual extraction in Figure 4(a). For both the

base model and the non-infringing model, we present the images generated with both target style and surrounding styles. We can see that the *extraction* operation successfully extract the target style from the base model while keeping the quality of images with the surrounding styles.

In Table 1, we present objective measures to assess the performance of the *extraction* operation in comparison to baseline methods. Our method demonstrates a notable improvement, with the KID metric increasing from 42 to 187 on target style compared to Concepts-Ablation (Kumari et al., 2023), which indicates better removal of the target style. In a comparative evaluation with the Erasing method (Gandikota et al., 2023), our approach achieves a reduction in LPIPS (Learned Perceptual Image Patch Similarity) from 0.21 to 0.14 when considering the impact on surrounding styles. This reduction indicates less damage to the surrounding artistic elements, justifying our method's efficacy in protecting the quality of the generated images with surrounding prompts.

Table 1: **Quantitative comparison with baselines in style transfer.** We compare the efficacy of similar methods under their metrics. Compared to Concepts-Ablation, ours removes target style more thoroughly, and compared to ESD, ours have less damage to surrounding styles.

| Metrics | Methods | Target style ↑ | Surrounding style ↓ |
|---|---|---|---|
| KID×$10^3$ | Extraction (Ours) | **187** | 32 |
| | Concepts-Ablation | 42 | 12 |
| LPIPS | Extraction (Ours) | 0.31 | **0.14** |
| | ESD (Gandikota et al., 2023) | 0.38 | 0.21 |

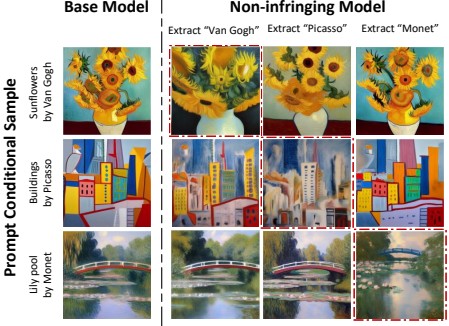

(a) Results of *extraction* in style transfer.

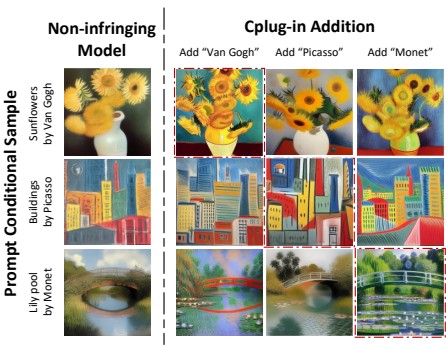

(b) Results of *combination* in style transfer

Figure 4: **Results in style transfer**. In Figure(a), We show samples from different non-infringing model in each column. Each model exhibits a deficiency in one style generation ability, with all other style generation capabilities remaining unaffected. In Figure(b), we present samples generated after adding certain ©Plug-ins in each column. Each of these ©Plug-ins serves to exclusively restore the generation of one particular style, while the generation of other styles continues to exhibit diminished performance.

**Combination.** We combine the above three ©Plug-ins to construct a non-infringing model devoid of these three artistic styles. We then individually add each style ©Plug-in to this model. Figure 4(b) shows that by using the combination method, the generative model can simultaneously remove multiple styles. Remarkably, ©Plug-in can restore its capability to generate artworks of target style without contravening copyright restrictions associated with other artistic styles.

### 3.3 EXTRACTION AND COMBINATION OF CARTOON IPS

For IP recreation, we show results on *Extraction* and *Combination*. Figure 5 shows three IP characters *extraction*: Mickey, R2D2 and Snoopy. It performs well on all of them, extracting the given IP

without disturbing the generation of other IPs. Table 2 quantifies extraction effect in IP recreation. We can increase the KID of the target IP by approximately 2.6 times while keeping the KID of the surrounding IP approximately unchanged.

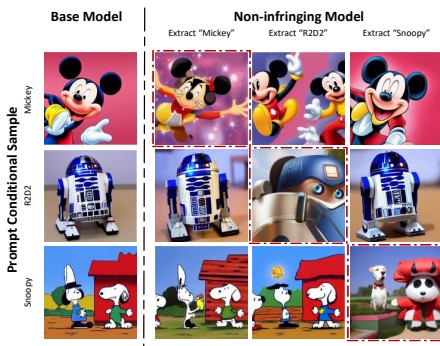

Figure 6: **IP *addition* within a single image.** We can add ©Plug-in to generate Mickey or Vader in a single image or add combined ©Plug-in to generate both.

Figure 5: **Results of *extraction* in IP recreation.** Each column of images is generated by a distinct non-infringing model. We can extract any individual IP of Mickey, R2D2, and Vader without affecting the generation of other IPs.

Table 2: **Quantitative comparison in IP recreation.** we increase the KID of target IP about 2.6 times compared with Concepts-Ablation, while keeping the KID of surrounding IP on par.

| Metrics | Methods | Target IP ↑ | Surrounding IP ↓ |
|---------|---------|-------------|-------------------|
| KID×10³ | Extraction (Ours) | **131** | 17 |
|         | Concepts-Ablation | 50 | 15 |

Furthermore, we illustrate *combination* and *addition* of various IP in a single image, as exemplified in Figure 6. Subsequent to the *combination* step, the non-infringing model's capability to generate either Mickey Mouse or Darth Vader-themed images is removed. Upon the addition of the corresponding ©Plug-in, the model is once again empowered to produce IP-related content, albeit exclusively within the domain of the added IP. However, the capability to generate the other IP remains disabled. When adding the combined ©Plug-in, the model successfully recovers the capability to generate both IPs in one image.

## 4  RELATED WORK

In order to position our work in the vast literature, we review related work through two perspectives: scope and technique. It is worthy to note that some of the literature touch both sides and we organize them in a way most related to ours. Due to the space limit, we review the technique-related literature here and defer the rest to Appendix B.

Our *extraction* operation is closely related with the *concept removal* for generative models. Gandikota et al. (2023); Kumari et al. (2023) remove target concepts by matching the generation distribution of contexts with target concepts and that of contexts without target concepts. Zhang et al. (2023a) forget target concepts by minimizing the cross attention of target concepts with that of target images. Heng & Soh (2023) leverage the reverse process of continual learning to promote the controllable forgetting of target contents in deep generative models.

We note that negative sampling (Ho & Salimans, 2022) can also prevent generating certain concepts. Specifically, end users can set conditional context and negative context to guide the diffusion process to generate images conforming the conditional context while being far away from the negative context. Only negative sampling cannot stop copyright infringing generation because the contexts are set freely and adversarially by end users.

In contrast, for a specific copyrighted concept, our *extraction* operation takes an "inverse LoRA" approach to disentangle the base model into two part: non-infringing base model and the plug-in LoRA component for copyrighted concept. Specifically, we use negative sampling to generate non-infringing images, which serves as training data for copyright plug-in. From the aspect of parameter efficient fine-tuning, our paper is related with literature (Alaluf et al., 2022; Ruiz et al., 2023; Gal et al., 2022; Hu et al., 2021; Huang et al., 2023).

Our *combination* operation is related with the widely studied "knowledge distillation" (Liang et al., 2023; Lopes et al., 2017; Sun et al., 2019; Hinton et al., 2015; Fang et al., 2019), but entails large difference from previous work. We combine multiple copyright plug-ins that are LoRA components for different targets, and we take data free approach due to practical constraint.

## 5 DISCUSSION, OPEN QUESTIONS AND LIMITATIONS

There are increasing concerns that generative AI models may generate copyright infringing content, which is escalated as the state-of-the-art models keep improving the quality of generated images while cannot attribute proper credits to the original data in the training set. We provide a solution "©Plug-in Market"to mitigate such concerns in society, which is motivated by the purpose of copyright law. We demonstrate that the copyrighted data can be integrated into LoRA plug-ins of the base model, with which one can easily track the usage and fairly attribute the reward.

One challenging question for this framework is how to efficiently manage the plug-ins when their number becomes large, so that the end user can easily find the most appropriate plug-ins for certain generation. Moreover, when the base model is upgraded, the pool of plug-ins need to be retrained, which adds huge cost. Therefore, some backward compatibility needs to take into account. One limitation of current paper is that the performance of the non-infringing model may degrade if conducting too many extraction operations, and the influence is not thoroughly evaluated.

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

Copyright. https://en.wikipedia.org/wiki/Copyright.

diffusers. URL https://github.com/huggingface/diffusers/issues/2613.

AI Art Generators Spark Multiple Copyright Lawsuits. https://www.hollywoodreporter.com.

Ryan Abbott and Elizabeth Rothman. Disrupting creativity: Copyright law in the age of generative artificial intelligence. *Florida Law Review, Forthcoming*, 2022.

Yuval Alaluf, Omer Tov, Ron Mokady, Rinon Gal, and Amit Bermano. Hyperstyle: Stylegan inversion with hypernetworks for real image editing. In *Proceedings of the IEEE/CVF conference on computer Vision and pattern recognition*, pp. 18511–18521, 2022.

Mikołaj Bińkowski, Danica J Sutherland, Michael Arbel, and Arthur Gretton. Demystifying MMD GANs. In *International Conference on Learning Representations*, 2018.

Olivier Bousquet, Roi Livni, and Shay Moran. Synthetic data generators–sequential and private. *Advances in Neural Information Processing Systems*, 33:7114–7124, 2020.

Tom Brown, Benjamin Mann, Nick Ryder, Melanie Subbiah, Jared D Kaplan, Prafulla Dhariwal, Arvind Neelakantan, Pranav Shyam, Girish Sastry, Amanda Askell, Sandhini Agarwal, Ariel Herbert-Voss, Gretchen Krueger, Tom Henighan, Rewon Child, Aditya Ramesh, Daniel Ziegler, Jeffrey Wu, Clemens Winter, Chris Hesse, Mark Chen, Eric Sigler, Mateusz Litwin, Scott Gray, Benjamin Chess, Jack Clark, Christopher Berner, Sam McCandlish, Alec Radford, Ilya Sutskever, and Dario Amodei. Language models are few-shot learners. In *Advances in Neural Information Processing Systems*, pp. 1877–1901. Curran Associates, Inc.

Nicolas Carlini, Jamie Hayes, Milad Nasr, Matthew Jagielski, Vikash Sehwag, Florian Tramer, Borja Balle, Daphne Ippolito, and Eric Wallace. Extracting training data from diffusion models. In *32nd USENIX Security Symposium (USENIX Security 23)*, pp. 5253–5270, 2023.

Ali Edalati, Marzieh Tahaei, Ivan Kobyzev, Vahid Partovi Nia, James J Clark, and Mehdi Rezagholizadeh. Krona: Parameter efficient tuning with kronecker adapter. *arXiv preprint arXiv:2212.10650*, 2022.

Niva Elkin-Koren, Uri Hacohen, Roi Livni, and Shay Moran. Can copyright be reduced to privacy? *arXiv preprint arXiv:2305.14822*, 2023.

Gongfan Fang, Jie Song, Chengchao Shen, Xinchao Wang, Da Chen, and Mingli Song. Data-free adversarial distillation. *arXiv preprint arXiv:1912.11006*, 2019.

Vitaly Feldman. Does learning require memorization? a short tale about a long tail. In *Annual ACM SIGACT Symposium on Theory of Computing*, 2020.

Giorgio Franceschelli and Mirco Musolesi. Copyright in generative deep learning. *Data & Policy*, 4:e17, 2022.

Rinon Gal, Yuval Alaluf, Yuval Atzmon, Or Patashnik, Amit H Bermano, Gal Chechik, and Daniel Cohen-Or. An image is worth one word: Personalizing text-to-image generation using textual inversion. *arXiv preprint arXiv:2208.01618*, 2022.

Rohit Gandikota, Joanna Materzynska, Jaden Fiotto-Kaufman, and David Bau. Erasing concepts from diffusion models. *arXiv preprint arXiv:2303.07345*, 2023.

Alvin Heng and Harold Soh. Selective amnesia: A continual learning approach to forgetting in deep generative models. *arXiv preprint arXiv:2305.10120*, 2023.

Martin Heusel, Hubert Ramsauer, Thomas Unterthiner, Bernhard Nessler, and Sepp Hochreiter. Gans trained by a two time-scale update rule converge to a local nash equilibrium. *Advances in neural information processing systems*, 30, 2017.

Geoffrey Hinton, Oriol Vinyals, and Jeff Dean. Distilling the knowledge in a neural network. *arXiv preprint arXiv:1503.02531*, 2015.

Jonathan Ho and Tim Salimans. Classifier-free diffusion guidance. *arXiv:2207.12598*, 2022.

Jonathan Ho, Ajay Jain, and Pieter Abbeel. Denoising diffusion probabilistic models. *Advances in neural information processing systems*, 33:6840–6851, 2020.

Edward J Hu, Yelong Shen, Phillip Wallis, Zeyuan Allen-Zhu, Yuanzhi Li, Shean Wang, Lu Wang, and Weizhu Chen. Lora: Low-rank adaptation of large language models. *arXiv preprint arXiv:2106.09685*, 2021.

Chengsong Huang, Qian Liu, Bill Yuchen Lin, Tianyu Pang, Chao Du, and Min Lin. Lorahub: Efficient cross-task generalization via dynamic lora composition. *arXiv preprint arXiv:2307.13269*, 2023.

Nam Hyeon-Woo, Moon Ye-Bin, and Tae-Hyun Oh. Fedpara: Low-rank hadamard product for communication-efficient federated learning. In *International Conference on Learning Representations*, 2021.

Diederik P Kingma and Max Welling. Auto-encoding variational bayes. *arXiv preprint arXiv:1312.6114*, 2013.

Nupur Kumari, Bingliang Zhang, Sheng-Yu Wang, Eli Shechtman, Richard Zhang, and Jun-Yan Zhu. Ablating concepts in text-to-image diffusion models. *arXiv preprint arXiv:2303.13516*, 2023.

Brian Lester, Rami Al-Rfou, and Noah Constant. The power of scale for parameter-efficient prompt tuning. *arXiv preprint arXiv:2104.08691*, 2021.

Hanlin Li, Brent Hecht, and Stevie Chancellor. All that's happening behind the scenes: Putting the spotlight on volunteer moderator labor in reddit. In *Proceedings of the International AAAI Conference on Web and Social Media*, volume 16, pp. 584–595, 2022a.

Hanlin Li, Brent Hecht, and Stevie Chancellor. Measuring the monetary value of online volunteer work. In *Proceedings of the International AAAI Conference on Web and Social Media*, volume 16, pp. 596–606, 2022b.

Xiang Lisa Li and Percy Liang. Prefix-tuning: Optimizing continuous prompts for generation. *arXiv preprint arXiv:2101.00190*, 2021.

Chen Liang, Simiao Zuo, Qingru Zhang, Pengcheng He, Weizhu Chen, and Tuo Zhao. Less is more: Task-aware layer-wise distillation for language model compression. In *International Conference on Machine Learning*, pp. 20852–20867. PMLR, 2023.

Tsung-Yi Lin, Michael Maire, Serge Belongie, Lubomir Bourdev, Ross Girshick, James Hays, Pietro Perona, Deva Ramanan, C. Lawrence Zitnick, and Piotr Dollár. Microsoft coco: Common objects in context, 2015.

Yulong Liu, Guibo Zhu, Bin Zhu, Qi Song, Guojing Ge, Haoran Chen, GuanHui Qiao, Ru Peng, Lingxiang Wu, and Jinqiao Wang. Taisu: A 166m large-scale high-quality dataset for chinese vision-language pre-training. *Advances in Neural Information Processing Systems*, 35:16705–16717, 2022.

Raphael Gontijo Lopes, Stefano Fenu, and Thad Starner. Data-free knowledge distillation for deep neural networks. *arXiv preprint arXiv:1710.07535*, 2017.

OpenAI. Dall·E 3. `https://openai.com/dall-e-3`, 2023.

Alec Radford, Jong Wook Kim, Chris Hallacy, Aditya Ramesh, Gabriel Goh, Sandhini Agarwal, Girish Sastry, Amanda Askell, Pamela Mishkin, Jack Clark, Gretchen Krueger, and Ilya Sutskever. Learning transferable visual models from natural language supervision. In *International conference on machine learning*, pp. 8748–8763. PMLR, 2021.

Evani Radiya-Dixit, Sanghyun Hong, Nicholas Carlini, and Florian Tramer. Data poisoning won't save you from facial recognition. In *International Conference on Learning Representations*, 2021.

Aditya Ramesh, Mikhail Pavlov, Gabriel Goh, Scott Gray, Chelsea Voss, Alec Radford, Mark Chen, and Ilya Sutskever. Zero-shot text-to-image generation. In *International Conference on Machine Learning*, pp. 8821–8831. PMLR, 2021.

Aditya Ramesh, Prafulla Dhariwal, Alex Nichol, Casey Chu, and Mark Chen. Hierarchical text-conditional image generation with clip latents. *arXiv preprint arXiv:2204.06125*, 1(2):3, 2022.

Javier Rando, Daniel Paleka, David Lindner, Lennart Heim, and Florian Tramer. Red-teaming the stable diffusion safety filter. In *NeurIPS ML Safety Workshop*, 2022.

Robin Rombach, Andreas Blattmann, Dominik Lorenz, Patrick Esser, and Björn Ommer. High-resolution image synthesis with latent diffusion models. In *Proceedings of the IEEE/CVF Conference on Computer Vision and Pattern Recognition (CVPR)*, pp. 10684–10695, June 2022a.

Robin Rombach, Andreas Blattmann, Dominik Lorenz, Patrick Esser, and Björn Ommer. High-resolution image synthesis with latent diffusion models. In *Proceedings of the IEEE/CVF conference on computer vision and pattern recognition*, pp. 10684–10695, 2022b.

Olaf Ronneberger, Philipp Fischer, and Thomas Brox. U-net: Convolutional networks for biomedical image segmentation. In *Medical Image Computing and Computer-Assisted Intervention–MICCAI 2015: 18th International Conference, Munich, Germany, October 5-9, 2015, Proceedings, Part III 18*, pp. 234–241. Springer, 2015.

Nataniel Ruiz, Yuanzhen Li, Varun Jampani, Yael Pritch, Michael Rubinstein, and Kfir Aberman. Dreambooth: Fine tuning text-to-image diffusion models for subject-driven generation. In *Proceedings of the IEEE/CVF Conference on Computer Vision and Pattern Recognition*, pp. 22500–22510, 2023.

Patrick Schramowski, Manuel Brack, Björn Deiseroth, and Kristian Kersting. Safe latent diffusion: Mitigating inappropriate degeneration in diffusion models. In *Proceedings of the IEEE Conference on Computer Vision and Pattern Recognition (CVPR)*, 2023.

Christoph Schuhmann, Romain Beaumont, Richard Vencu, Cade Gordon, Ross Wightman, Mehdi Cherti, Theo Coombes, Aarush Katta, Clayton Mullis, Mitchell Wortsman, et al. Laion-5b: An open large-scale dataset for training next generation image-text models. *Advances in Neural Information Processing Systems*, 35:25278–25294, 2022.

Shawn Shan, Jenna Cryan, Emily Wenger, Haitao Zheng, Rana Hanocka, and Ben Y Zhao. Glaze: Protecting artists from style mimicry by text-to-image models. *arXiv preprint arXiv:2302.04222*, 2023.

Jascha Sohl-Dickstein, Eric Weiss, Niru Maheswaranathan, and Surya Ganguli. Deep unsupervised learning using nonequilibrium thermodynamics. In *International conference on machine learning*, pp. 2256–2265. PMLR, 2015.

Yang Song, Jascha Sohl-Dickstein, Diederik P Kingma, Abhishek Kumar, Stefano Ermon, and Ben Poole. Score-based generative modeling through stochastic differential equations. In *International Conference on Learning Representations*, 2020.

Siqi Sun, Yu Cheng, Zhe Gan, and Jingjing Liu. Patient knowledge distillation for bert model compression. In *Proceedings of the 2019 Conference on Empirical Methods in Natural Language Processing and the 9th International Joint Conference on Natural Language Processing (EMNLP-IJCNLP)*, pp. 4323–4332, 2019.

Anton Troynikov. Stable Attribution. https://www.stableattribution.com, 2023.

Nicholas Vincent and Brent Hecht. A deeper investigation of the importance of wikipedia links to search engine results. *Proceedings of the ACM on Human-Computer Interaction*, 5(CSCW1): 1–15, 2021.

Nicholas Vincent, Hanlin Li, Nicole Tilly, Stevie Chancellor, and Brent Hecht. Data leverage: A framework for empowering the public in its relationship with technology companies. In *Proceedings of the 2021 ACM Conference on Fairness, Accountability, and Transparency*, pp. 215–227, 2021.

Nikhil Vyas, Sham Kakade, and Boaz Barak. Provable copyright protection for generative models. *arXiv preprint arXiv:2302.10870*, 2023.

Eric Zhang, Kai Wang, Xingqian Xu, Zhangyang Wang, and Humphrey Shi. Forget-me-not: Learning to forget in text-to-image diffusion models. *arXiv preprint arXiv:2303.17591*, 2023a.

Lvmin Zhang, Anyi Rao, and Maneesh Agrawala. Adding conditional control to text-to-image diffusion models, 2023b.

Richard Zhang, Phillip Isola, Alexei A Efros, Eli Shechtman, and Oliver Wang. The unreasonable effectiveness of deep features as a perceptual metric. In *CVPR*, 2018.

Haonan Zhong, Jiamin Chang, Ziyue Yang, Tingmin Wu, Pathum Chamikara Mahawaga Arachchige, Chehara Pathmabandu, and Minhui Xue. Copyright protection and accountability of generative ai: Attack, watermarking and attribution. In *Companion Proceedings of the ACM Web Conference 2023*, pp. 94–98, 2023.

