## A   INTERMEDIATE RESULTS OF EXTRACTION

We present some intermediate results of *extraction*. It becomes evident that the non-infringing model's output is predominantly limited to low-quality noise following the unlearning phase. However, upon proceeding to the memorization phase, the model's overall image generation capability is reinstated, but it cannot yet generate artwork in the style of Picasso.

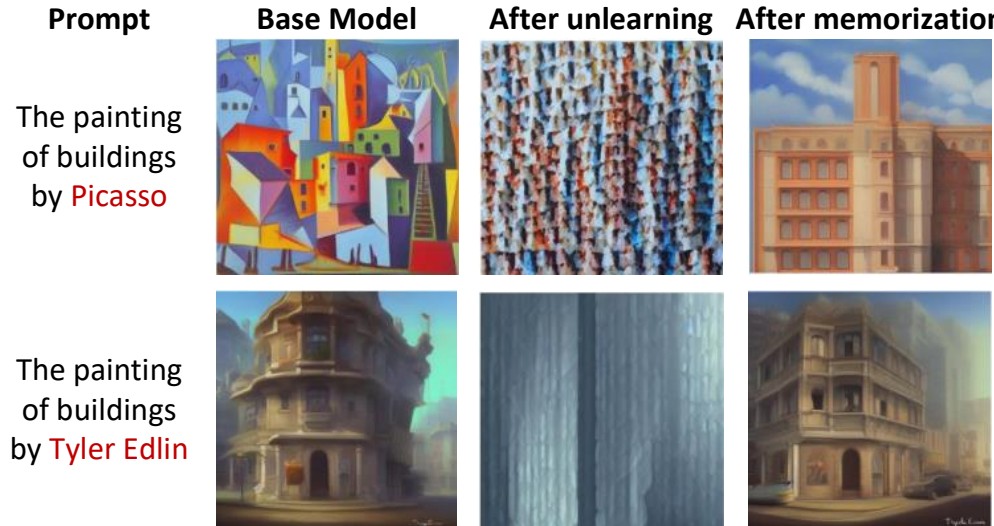

Figure 7: **Intermediate results of *extraction*.** After unlearning, the non-infringing model's generative abilities become significantly limited, predominantly manifesting as the production of noise. After memorization, the generation prowess is rejuvenated, but due to the absence of learning Picasso-style images, the model remains unable to generate artwork in the style of Picasso.

## B   MORE RELATED WORKS

### B.1   SCOPE RELATED: COPYRIGHT, DATA CONTRIBUTION AND CREDIT ATTRIBUTION

Recent text-to-image generative models are trained with large scale datasets (Schuhmann et al., 2022; Liu et al., 2022), which cannot be guaranteed free of copyrighted data. At the same time, the state-of-the-art models are capable of generating high-quality and valuable creative images comparable to human creators or even memorizing the data points in the training set (Carlini et al., 2023), which arouses copyright concerns about the training data and brings anxiety to the artist community.

Numerous efforts have been made for copyright protection of training data (Zhong et al., 2023). A direct approach is removing the copyrighted images from the training set, which may involve cumbersome cost due to the size of the training sets and may significantly degrade the model performance (Feldman, 2020). Another direct approach is post filtering, refusing to generate images with copyrighted concepts, e.g., Schramowski et al. (2023) proposes *Safe Latent Diffusion* to guide latent representation away from target concepts in the inference process, which nonetheless can be bypassed by a user with access to the model (Rando et al., 2022). As an example, OpenAI Dall·E3 (OpenAI, 2023) declines requests for generating an image in the style of a living artist and promises that creators can also opt their images out from training of future image generation models. Many papers discuss the idea of concept removal, which will be reviewed in later section.

Shan et al. (2023) propose *Image Cloaking* that suggests adding adversarial perturbations before posting artistic works on the internet so as to make them unlearnable for machine learning model, which has been pointed out to be hard to defend against future learning algorithms (Radiya-Dixit et al., 2021).

Theoretically, Bousquet et al. (2020); Elkin-Koren et al. (2023) connect the copyright protection of training data with the concept of differential privacy and discuss their subtle differences. Vyas et al. (2023) further formulate the copyright problem with a *near free access* (NAF) notion to bound the distance of the generative distributions of the models trained with and without the copyrighted data.

Our paper distinguishes largely from all previous works as we do not try to prohibit generating copyrighted concepts but instead we introduce a copyright market for the generative model to reward the copyright owners with fairness and transparency. From this aspect, our paper is also related with literature of monetizing the training data (Vincent & Hecht, 2021; Vincent et al., 2021; Li et al., 2022b;a) or attributing credits for the generative contents (Troynikov, 2023), but we establish a very distinct way to reward the authorship.

Heated discussion is also around the copyright for AI generated art work Franceschelli & Musolesi (2022); Abbott & Rothman (2022). The Review Board of the United States Copyright Office has recently refused the copyright registration of a two-dimensional AI generated artwork entitled "A Recent Entrance to Paradise". However, Abbott & Rothman (2022) argues for giving the copyright to AI generated works, which will encourage people to develop and use creative AI, promote transparency and eventually benefit the public interest.

## C    RESULT OF ORDINARY OBJECTS GENERATION

We evaluated the impact of *extraction* on ordinary objects generation. We selected 5000 textual captions from the validation set in MS-COCO (Lin et al., 2015) as prompts, and then generated 5000 images using SD1.5 and the non-infringing model that extracts R2D2 and Picasso, respectively. we randomly displayed several images in Figure 8. To demonstrate the effect, we also generated results for concept-ablation and ESD on MS-COCO, respectively.

As shown in Table3, we have calculated some quantitative metrics like FID and KID. The results indicate that the *extraction* has little impact on image generation in MS-COCO. All methods have almost no weakening in generating ordinary objects. We note that Concept-Ablation (Kumari et al., 2023) only releases the checkpoint of ablating "R2D2" and ESD (Gandikota et al., 2023) only releases the checkpoint of erasing "Picasso". Therefore we compare with them respectively by using their own checkpoints.

Table 3: **Quantitative results on MS-COCO.** We quantified the impact of *extraction* and other removal techniques on the generation of ordinary objects. The calculated FID and KID metrics for removing the IP character R2D2 are presented in the upper two rows, while that for removing the Picasso style are displayed in the lower two rows. The findings reveal a nearly similarity between these two sets of images, whether the extraction pertained to R2D2 or Picasso.

| Method | FID $\downarrow$ | KID$\times 10^3 \downarrow$ |
|---|---|---|
| *Extract R2D2* | 20.55 | 2.36 |
| *Ablate R2D2* | 18.97 | 1.34 |
| | | |
| *Extract Picasso* | 24.04 | 2.83 |
| *Erase Picasso* | 25.20 | 3.39 |

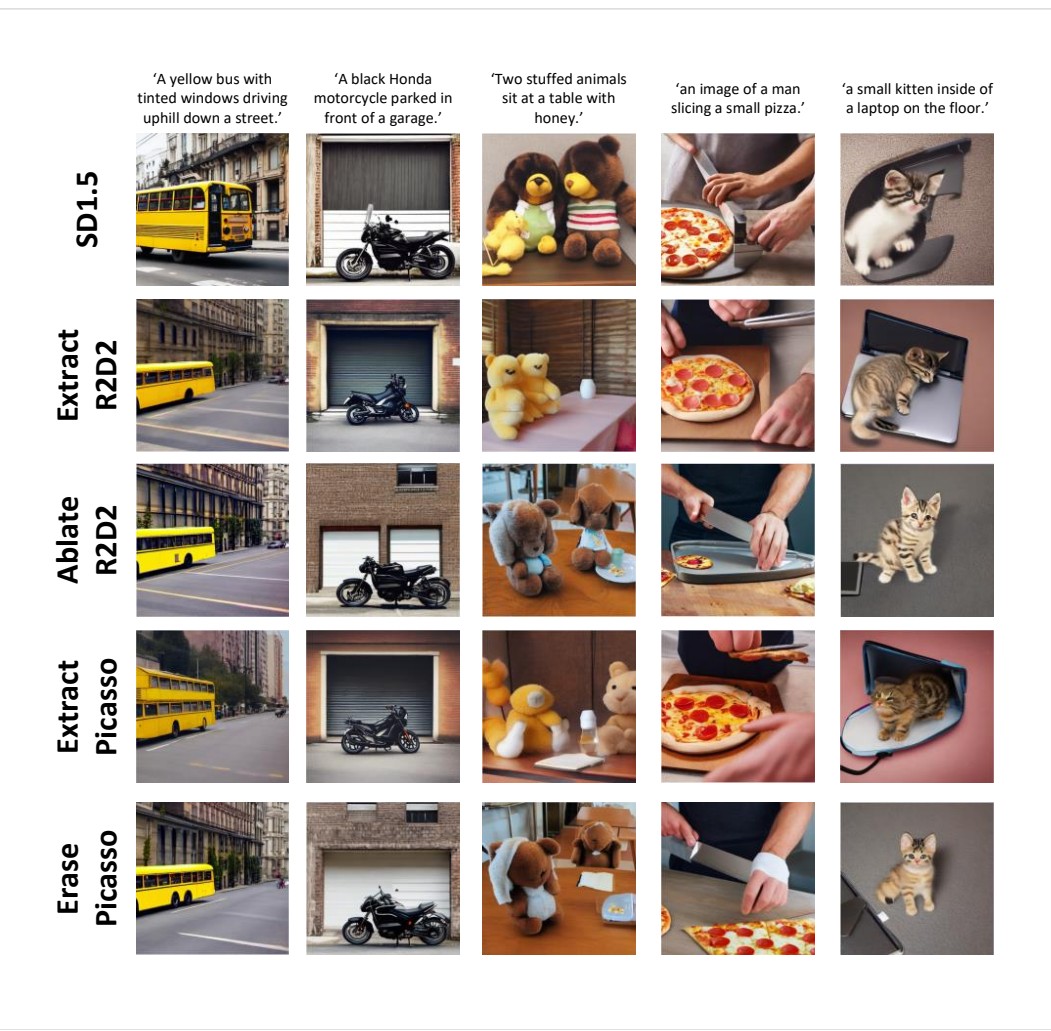

Figure 8: **Ordinary objects generation after *extraction*.** Row 1 displays images generated by Stable Diffusion V-1.5. Rows 2 and 3 illustrate images generated subsequent to the removal of the IP character R2D2, while Rows 4 and 5 showcase images generated after the elimination of Picasso's style. Rows 3 and 5 serve as the baseline, representing concept-ablation and ESD, respectively. Notably, after the *extraction* of R2D2 and Picasso, the non-infringing model retains the capability to generate commonplace objects sourced from the MS-COCO dataset. (Lin et al., 2015).