# OpenReview forum: "Copyright Plug-in Market for The Text-to-Image Copyright Protection"
_ICLR.cc/2024/Conference — Submitted to ICLR 2024_

### Official Review · Reviewer_5G9e · 2023-10-31

**Soundness:** 2 fair
**Presentation:** 3 good
**Contribution:** 3 good
**Rating:** 5
**Confidence:** 4

**Summary:**

The paper addresses the emergent and crucial issue of model unlearning. Notably, many contemporary generative models face issues of infringement, as some artists are reluctant to have their work used in training such models. The authors introduce a dual-phase approach: the LoRA fine-tuning stage for concept elimination, followed by a memory stage to enhance generation quality for non-infringing images. The paper mainly evaluates the proposed method on well-defined concepts such as Picasso and Mickey.

**Strengths:**

1. The topic is novel and the authors state the importance of the topic very clearly.
2. The proposed method is interesting. It is good to see that the LoRA module could be used for concept extraction.

**Weaknesses:**

1. The evaluation is weak. The current results cannot be used to fully support the effectiveness of the proposed method. Actually, the evaluation of this kind of task is also very important.

**Questions:**

1. With the concept of removed model weights, what about the impact on the FID on larger datasets such as Laion and COCO?
2. How to forget those concepts that are not well defined by text words?

---

> ### Author Response · Authors · 2023-11-23
> **Response with more evaluation and experiments**
>
> Thank you for the constructive suggestions. We have conducted more evaluation on diverse data sets and updated the paper in Appendix C.
>
> **Q1:** More evaluations are needed.
>
> **A1:** We conduct *extraction* on MS-COCO 2017. We sample 5000 images by Stable Diffusion Model (SDM) 1.5 and non-infringing models after extracting R2D2 and Picasso, respectively. All results are displayed in Appendix C Table 3.
>
> We directly use the model checkpoint in the paper and obtain the following number: the FID  20.55 and the KID ($\times 10^3$)  2.36 for extracting R2D2, and the FID 24.04 and the KID ($\times 10^3$) 2.83 for extracting Picasso. It indicates that the distribution of images generated by the base model and the model after extraction is very similar. The capability of the model after extraction keeps nearly uncharged (see Figure 8 in Appendix C).
>
> **Q2:** How to forget those concepts that are not well defined by text words?
>
> **A2:** Currently even the base models have difficulty to learn a concept that cannot be described by text words. For those concepts, it is also difficult for users to generate them in a controllable way. It is an interesting future direction to clearly define and identify cases for such infringements and then design techniques for protection, which deserves independent studies.
>
> Please let us know if you need any further clarifications. If our answers address your concerns, please consider raising the score. We are indeed excited by this work, and we believe that it provides a brand new way of copyright protection with technical feasibility. It is beneficial to convey these ideas to a broader AI+Society audience on a venue like ICLR.

---

### Official Review · Reviewer_Pa89 · 2023-10-31

**Soundness:** 3 good
**Presentation:** 1 poor
**Contribution:** 2 fair
**Rating:** 3
**Confidence:** 2

**Summary:**

This paper presents a conceptual framework to facilitate proper credit attribution in the text-to-image procedure and enable digital copyright protection. Three operations are proposed in this framework: addition, extraction, and combination. These basic operations are expected to give good incentives to each participant in the market and enable enough flexibility to thrive in the market. Then, an inverse LoRA approach is adopted to instantiate the extraction operation and a data-free layer-wise distillation is used to combine the multiple extractions or additions. The experiments conducted for style transfer and cartoon IP recreation demonstrate the effectiveness to extent of the proposed method.

**Strengths:**

1. This paper pays attention to a critical issue: digital copyright protection and credit attribution in the text-to-image generation procedure.
2. The idea and design are preliminary and somehow make sense.

**Weaknesses:**

1. The paper is poorly written and can be largely improved. As illustrated in the abstract, the framework aims to address the tension between the users, the content creators and the generative models. However, there are no figures, workflow or formulation that strictly define and describe how the proposed model interacts with these roles.
2. Most components to construct the model are widely adopted. What are the real merits of the work?
3. The design of the three operations feels quite ad hoc, and there is no explanation as to why these three operations precisely address the problem. Whether these three operations form a complete set or can represent all possible operations remains unanswered. It seems that some potential points have been mentioned rather cursorily, and there hasn't been a comprehensive exploration of this issue.

**Questions:**

Please see the weakness.

---

> ### Author Response · Authors · 2023-11-23
> **Response with revisions, clarifications and justification of technical novelty**
>
> Thank you for the suggestions. We address the weakness points one by one.
>
> **Q1:**  No figures, workflow or formulation that strictly define and describe how the proposed model interacts with these roles.
>
>
> **A1:** We have carefully revised the paper according to the comments. Specifically, we add a new paragraph (marked in blue) and a new figure (Figure 1) to illustrate the roles of participants in the market. Within the Copyright Plug-in Market, the **base model owner** (like *Stability AI*) acts as a store of copyright plug-ins, **copyright holders** (like *artists*) can register their copyright works as plug-ins and get reward from the usage of copyright plug-ins, while **end users** pay for generating images of  copyrighted concepts with corresponding plug-ins. This framework gives good incentives to all participants: the copyright holders are well compensated for creating new works, the end users pay for using copyrighted plug-ins and avoid being accused of copyright infringement in their own creations, and the base model owner makes profits for the plug-in registration and usage, as illustrated in Figure 1.
>
>
>
> **Q2:** Most components to construct the model are widely adopted. What are the real merits of the work?
>
> **A2:** First and foremost, we are taking steps to tackle a challenging problem of copyright infringement in the generative AI era. We propose to  protect copyright through an accurate credit attribution system, which is instantiated as a copyright market.  We note that previous work (Vyas et al., 2023 or Troynikov, 2023) either involves retraining the base model (of very high cost) or does not have a feasible technical solution for efficient copyright protection. Our solution is promising, technically feasible and has not been studied before.
>
> Second, as we stated in the paper, *addition* via LoRA is ready to use in public. However, currently there is no off-the-shelf solution for *extraction* of copyrighted work from a base model to a light-weight parameter. Our ''inverse LoRA'' approach is novel, effective and matching the need for building the copyright market. Our proposal of ''data-free layer-wise distillation'' is also new for combing multiple  LoRAs and perfectly matches the need of the market. These three operations can all be easily implemented with little fine-tuning effort or data requirement.
>
> **Q3:** The design of the three operations feels quite ad hoc, and there is no explanation as to why these three operations precisely address the problem.
>
> **A3:** Our proposal of three basic operations *addition, extraction* and *combination*, encompassing the fundamental/natural functionalities required for an effective and flexible market. However, we cannot argue  completeness of the operations given the emerging requirements and scenarios in the future. Other techniques may be also needed to enhance market security and availability, e.g., watermark technology can be used to regulate user behavior. This is out of the scope of the paper, and should better be investigated separately.
>
> If our answers change your evaluation of the paper, please consider raising the score. We believe that our paper has an important message for the copyright protection in the generative AI era, and venues like ICLR are an good place to convey this.

---

### Official Review · Reviewer_3Us4 · 2023-11-02

**Soundness:** 3 good
**Presentation:** 2 fair
**Contribution:** 3 good
**Rating:** 6
**Confidence:** 3

**Summary:**

The paper addresses a very important problem with Gen AI models, copyright infringing. Specifically, the work proposes a new framework for fair and transparent attribution in T2I  models to solve the copyright concern, with three basic operations addition, extraction and combination. A key novelty of the work is the proposal of Inverse LoRA algorithm to extract copyrighted concepts from the base model. Experiments and results demonstrate the utility of the proposed framework.

**Strengths:**

I applaud authors for attacking an important and challenging problem of copyright infringing. I thoroughly enjoyed reading the paper and really liked the overall framework (with some concerns though as highlighted in weaknesses) and believe the work could have high impact use cases, especially in industrial applications of genAI. The results provided in the paper, though limited, look great.

**Weaknesses:**

I do not find any major weaknesses with this work. However, I feel the paper could be further improved with more qualitative results for extraction, which is the key novelty of the work. I find the proposed method to be not very intuitive (section 2.2.1 and 2.2.2) and struggle to understand concretely how exactly it is able to achieve the extraction. Given limited quantitative results and comparisons, it would be helpful to see a diverse set of qualitative results. Also, I see that the paper mentions human evaluation in page 7, but doesn’t present any human eval results. I feel human eval results further help in justifying the proposed framework.

**Questions:**

Please see weaknesses.

---

> ### Author Response · Authors · 2023-11-23
> **Response with clarification and new experiments on more datasets**
>
> Thank you for the encouraging words and we do share the feeling on the importance of copyright infringing problem. We try to provide a simple and  feasible solution that could benefit participants in the society. In the following, we address the questions one by one.
>
>
> **Q1:** How does the extraction work intuitively?
>
> **A1:** Intuitively, the extraction can be thought as a reverse process of  ''adding LoRA for generating a certain concept''.  Essentially, we first learn a LoRA $w_L$ with respect to the infringing base model $w$ by aligning the copyrighted data (the image of ''the painting of buildings by Picasso'') with the text prompt  ''the painting of buildings''. As a result, the model $w+w_L$ can generate images with ''Picasso'' style even when the prompt does not contain ``Picasso'' text, which indicates that $w_L$ fully captures the information of generating ''Picasso'' style.
>
> Then we flip the sign of LoRA so that $w-w_L$ will be the non-infringing model parameters. This indeed unlearns the ''Picasso'' style from the base model (see Figure 7 in the Appendix). However, directly using $w-w_L$ as non-infringing model hurts the capability to generate images with surrounding and contextual text (''the painting of buildings'').
>
> We further tune the LoRA $\tilde{w}_L:=-w_L$ with pairs of images and contextual texts in the Memorization step and hope the LoRA can memorize the surrounding concepts well. Finally,  we obtain a non-infringing model $\tilde{w} = w+\tilde{w}_L$ and a \copyright plug-in given by $-\tilde{w}_L$. With the \copyright Plug-in, the model becomes the original SDM which can successfully generate artworks with the ''Picasso'' style.
>
> We have also revised Section 2.2 of the paper by including above discussions.
>
> **Q2:**  More quantification of the results.
>
> **A2:** In the paper we conducted comparison with algorithms for achieving similar goal, e.g., Erased Stable Diffusion (ESD) and Concept-Ablation with quantitative results. Specifically we use the metrics Kernel Inception Distance (KID) used in Concept-Ablation and and Learned Perceptual Image Patch Similarity (LPIPS) used in ESD, respectively,  to avoid any metric preference of the method. The results indicate that ours have surpassed or been on par with them even in their evaluation criteria.
>
> Nonetheless, human evaluation would be a golden standard of the metric. However, we did not conduct thorough human evaluation due to large cost of human evaluation and human subjective biases. We present several showcase examples (Figure 4,5,6,7) for reader's visual judgement.
>
> Moreover, following the reviewer's suggestion, we add more results of ordinary object generation with MS-COCO dataset in Appendix C. We evaluate the impact of *extraction* on ordinary objects generation. We select 5000 textual captions from the validation set in MS-COCO as prompts, and then generate 5000 images using SDM1.5 and the non-infringing model that extracts R2D2 and Picasso, respectively. we randomly display several images in Figure 8.
> As shown in Table 3, we calculate some quantitative metrics like FID and KID. The results indicate that the *extraction has little impact on image generation in MS-COCO*.
>
> Please let us know if you need any further clarifications. If our answers address your concerns, please do consider raising the score. We are indeed excited by this work, and we believe that it opens a brand new way of copyright protection with technical feasibility. It is beneficial to disseminate these ideas to a broader AI+Society audience on a venue like ICLR.

---

### Meta-Review · Area_Chair_XCRv · 2023-12-10

**Metareview:**

This paper proposes a method for transparent attribution in text-to-image genAI models via three operations based on fine-tuning Low-Rank Adaptation (LoRA) models on top of a generation: addition, extraction and combination. The method allows to extract existing copyrighted material via inverse LoRA. The authors evaluate the model on a set of 5000 prompts on Stable Diffusion 1.5 and non-infringing models.

Strenghts:
* All three reviewers applauded the important and challenging problem of copyright infringement
* The inverse LoRA approach was found innovative (5G9e)

Weaknesses:
* Reviewer 3Us4 asked for more qualitative results and evaluation
* The paper was found to lack in clarity (3Us4,Pa89) - the authors added some further explanation
* Quantitative results were deemed limited (3Us4,5G9e) - the reviewers added some more results by generating 5k MS-COCO images
* No human evaluation was provided (3Us4)

The scores (3, 5, 6) are well under the admission bar, and reviewers' comments suggest that the paper would gain in clarity and in more substantial evaluation. Based on this, I vote for rejecting the paper and wish the authors best of luck for revisions and resubmission at a different venue.

**Justification For Why Not Higher Score:**

Reviewers' comments suggest that the paper would gain in rewriting for clarity, and in more substantial evaluation.

**Justification For Why Not Lower Score:**

N/A

---

### Decision · Program_Chairs · 2024-01-16

Reject